# CO_2_-Accelerated Carbonation Modification for Recycled Coarse Aggregate with Various Original Concrete Strengths and Coarse Aggregate Sizes

**DOI:** 10.3390/ma17143567

**Published:** 2024-07-18

**Authors:** Wei Qin, Xinhui Fan, Xiaohui Jiang

**Affiliations:** 1School of Economics and Management, Chongqing Jiaotong University, 66 Xuefu Road, Nan’an District, Chongqing 400074, China; qinwei09011@126.com; 2National Engineering Research Center for Inland Waterway Regulation, School of River and Ocean Engineering, Chongqing Jiaotong University, 66 Xuefu Road, Nan’an District, Chongqing 400074, China

**Keywords:** CO_2_-accelerated carbonation, recycled coarse aggregate, original concrete strength, coarse aggregate size, property enhancement

## Abstract

The increasing demand for concrete reduces natural resources, such as sand and gravel, and also leads to a sharp increase in the amount of waste concrete produced. Due to the fact that the physical and mechanical properties of waste concrete made of recycled aggregates (RAs) differ greatly, it is difficult to use directly as a raw material for reinforced concrete (RC) components, which greatly restricts the popularization and application of RAs in actual projects. Utilizing the alkali aggregate properties of RAs to capture CO_2_ from industrial waste gases is an innovative way of enhancing their properties and promoting their application in real projects. However, the extent of the influence of original concrete strength (OCS) and coarse aggregate size (CAS) on the accelerated carbonation modification of RA is not clear, and a quantitative description is still required. For this purpose, accelerated carbonation tests on recycled coarse aggregate (RCA) samples under completely dry condition were carried out, and the variation laws for the physical property indicators of RCA samples before and after accelerated carbonation versus the OCS and CAS were revealed. Moreover, the influence degrees of the two factors, OCS and CAS, on the property enhancement of RCAs after accelerated carbonation were clarified, and the results of OCS and CAS corresponding to the best accelerated carbonation effects of RCAs were determined. By analyzing the micromorphology of RCA before and after accelerated carbonation, the reasons for property enhancement of RCAs with various OCSs and CASs under the best carbonation modifications were clarified. The findings will contribute to the development of basic theoretical research on accelerated carbonation modification of RA and have important scientific value.

## 1. Introduction

Concrete, as an important construction material, plays a crucial role in practical engineering [1,2]. Aggregates, as key raw materials for the preparation of concrete, serve as a skeleton and support the concrete, affecting many important properties of concrete materials [3,4,5]. With the acceleration of urbanization, large amounts of natural sand and gravel need to be extracted for the preparation of natural coarse aggregates (NCAs), decreasing the total stock of natural resources, such as sand and gravel [6]. At the same time, the amount of waste concrete is gradually increasing [7]. Sustainability and circularity have become key factors in developing the concrete industry, and incorporating more recycled materials and by-products is considered the way forward [8]. In this context, many scholars have crushed and screened waste concrete to produce recycled aggregates (RAs) [9,10,11] to alleviate the continuous exploitation of natural resources and reduce the negative impacts relating to the subsequent disposal of waste concrete. However, the properties of RAs are not as good as those of NCAs, seriously limiting the application of RAs in practical projects [12]. In addition, driven by the serious problem of increasing CO_2_ emissions in the concrete industry [13,14], utilizing the alkali aggregate properties of RAs to capture CO_2_ [15] while improving their own properties is considered an innovative way of effectively improving the properties of RAs [16,17,18] and realizing the application of modified RAs in actual projects [19,20], which has been a concerned of scholars in China and elsewhere [16,21].

It has been shown that the raw material properties of RAs (original concrete strength, coarse aggregate size, and initial moisture content) are important factors affecting the effectiveness of carbonation modifications [22,23,24]. The initial moisture content (IMC) of RAs is an important factor in determining the degree of enhancement in the overall quality and development of carbonized recycled aggregates (CRAs). The lower the IMC of RAs, the more they can capture CO_2_ and absorb water from the environment, and in a carbonation environment with high relative humidity, lowering the IMC of RAs can have a positive effect in terms of increasing their carbonation degree and enhancing their physical properties [25]. Ju et al. [26] conducted a more comprehensive comparative study on RAs in three IMC states: completely dry, untreated, and complete wetting. They concluded that completely dry conditions have the best effect on the carbonation modification of RAs.

It has been found that original concrete strength (OCS), as an important index used in the comprehensive evaluation of the properties of the original concrete, has a very significant effect on the macroscopic properties of RAs and CRAs [27,28,29]. Etxeberria et al. [30] found that the higher the OCS, the better the effect on the carbonation modification of RAs. Hyvert et al. [31] carried out RA carbonation with OCSs of C60 and C80 for 24 h and found a certain degree of increase in their apparent density; however, the improvement in the physical properties of RAs with OCSs of C30 and C45 was not obvious. The carbonizable material present in the RCA, combined with different OCSs and the overall degree of densification, affects the degree of improvement in RA performance. Therefore, for the use of high-OCS concrete preparations involving RA, despite the fact that greater hardened cement paste can be carbonized, the overall degree of compactness is also large, resulting in a carbonation medium that is difficult to penetrate; thus, the carbonation reaction is more difficult. Therefore, it is difficult to conclude that OCS is positively correlated with the carbonation effect of RAs. Research on the effect of OCS on the performance of RAs needs to be deepened. In addition, coarse aggregate size (CAS) significantly affects the carbonation degree of RAs, and the CO_2_ absorption rate and carbonation rate of RAs in the process of accelerated carbonation increase gradually with decreasing CAS [23,24,32,33]. Xuan et al. [34] used experimental studies to show that the CO_2_ absorption ratio of RCAs with a CAS of less than 5 mm was about 2.15%, and the absorption ratio of RCAs with CASs of 5–10 mm was 0.81%. Fang et al. [35] showed that the CO_2_ uptake of RAs with average CASs of 1.18 mm and 15 mm was about 54 g/kg and 27 g/kg, respectively. However, the effect of CAS on the degree of carbonation of RAs with different OCSs is not yet known, and the carbonation modification of RAs, when considering the joint effect of OCS and CAS, deserves in-depth study.

In summary, IMC, OCS, CAS, and other raw material factors of RAs will have varying degrees of impact on the performance improvement of carbonated recycled aggregate concrete (CRAC). Although it has been clarified that the carbonation modification effect is best when RCAs are completely dry, research on the quantitative impact of both OCS and CAS on the carbonation modification effect of RAs needs to be carried out. In addition, in a standard carbonation environment (ambient temperature, 20 ± 2 °C; relative humidity, 70 ± 5%; CO_2_ concentration, 20 ± 3%), the OCS and CAS of RAs that have the best effect on accelerating carbonation modification are not yet clear, and this issue still needs to be studied in depth. 

For this reason, in this paper, concrete specimens with OCSs of C30, C40, and C50 were subjected to crushing, sieving, drying, and other pretreatments to produce a sufficient amount of recycled coarse aggregate (RCA) specimens with various OCSs (C30, C40, and C50) and CASs (5~10 mm, 10~20 mm, and 20~25 mm). By carrying out accelerated carbonation tests on the RCA samples with a dry IMC, the change laws of the physical property indexes of the RCAs before and after accelerated carbonation with different OCSs and CASs are revealed, and the extent to which OCSs and CASs affect the accelerated carbonation modification of RCAs are clarified; in addition, the OCS and CAS results of RCAs corresponding to the best accelerated carbonation modification effects are determined. Through the micromorphological analysis of RCAs before and after accelerated carbonation, the reasons for the performance enhancement of RCAs with different OCSs and CASs under optimal carbonation modification conditions are clarified.

The highlights and innovations of this paper are as follows:(1)Carrying out accelerated carbonation tests on RCAs with various OCSs and CASs with a completely dry IMC revealed the changing rules of the physical property indexes of RCAs in terms of OCS and CAS before and after accelerated carbonation.(2)The influence of OCS and CAS on the accelerated carbonation modification of RCA was clarified, and the OCS and CAS of RCAs that had the optimal effect in terms of accelerated carbonation modification were identified.(3)By analyzing the microscopic morphology of RCA before and after accelerated carbonation, the reasons for the property enhancement of RCAs with various OCSs and CASs under optimal carbonation modification conditions were clarified.

## 2. Materials and Experiments

### 2.1. Raw Materials and Mix Proportions of Concrete for Producing the RCA

The raw materials used for the original concrete selected for the tests presented in this paper include PC.42.5R composite silicate cement (Chongqing Huaxin Yanjing Cement Co., Ltd, Chongqing, China) produced in the same batch (dry density of 3.1 × 10^3^ kg/m^3^), continuously graded coarse aggregate (nominal particle diameter of 5–25 mm), freshwater river sand (fineness modulus of 2.58), and ordinary tap water (density of 1000 kg/m^3^) [36].

According to the JTS 151-2011 standard [37], concrete cube specimens with OCSs of C30, C40, and C50 were prepared with geometrical dimensions of 100 × 100 × 100 mm^3^. The concrete mixes and their various OCSs are shown in Table 1. When casting the specimens, it was necessary to place the fresh concrete specimens and molds together in a standard curing box for initial curing for 24 h. Then, they were demolded and promptly placed in saturated Ca(OH)_2_ solution to continue curing for 28 days to test their compressive strength. The compressive strength tests of concrete specimens with different OCSs (3 *w*/*c*, a total of 9 specimens) were carried out by using a DYE-2000KN-type pressure tester (Cangzhou Keyu Road Industry, Cangzhou, China) and the results showed that the compressive strengths of the concrete specimens with *w*/*c* = 0.4, 0.5, and 0.6 were 49.9 MPa, 41.6 MPa, and 33.9 MPa, respectively, reaching the predetermined original concrete strengths of C50, C40, and C30.

Subsequently, these concrete specimens with different OCSs were crushed, sieved, and completely dried to obtain sufficient quantities of RCA specimens with various OCSs (C30, C40, and C50) and CASs (5–10 mm, 10–20 mm, and 20–25 mm), respectively, to be used in the subsequent CO_2_-accelerated carbonation modification test research.

### 2.2. RCA Sample Acquirements

In this test, a jaw crusher was used to crush concrete specimens with various OCSs, and the width of the crusher opening was pre-adjusted to generally control CAS of the crushed RCA. The original concrete specimens were crushed separately in the order of various OCSs of C30, C40, and C50, and they were promptly put in the preparation bags and sealed for the subsequent sieving process.

The RCA was sieved with three different CASs of 5–10 mm, 10–20 mm, and 20–25 mm, respectively. Wu et al. [25] and Ju et al. [26] showed that, under completely dry conditions, RCAs can rapidly absorb water and CO_2_ in the environment during accelerated carbonation processes, resulting in the best carbonation modification effects under this condition. In this paper, RCAs with different OCSs and CASs in completely dry conditions were selected as the research objects to investigate the effect of CO_2_-based accelerated carbonation on the carbonation modification degrees of the aforementioned RCA samples. Hence, it was necessary to transfer all of the RCA samples into a blower drying oven at a temperature of 105 ± 5 °C to dry until they reached a constant weight, and the mass of the RCA samples was subsequently weighed prior to further use.

As shown in Figure 1, the RCAs used for the CO_2_-accelerated carbonation modification study were all completely dry, and they included three OCSs (C30, C40, and C50) and three CASs (5–10 mm, 10–20 mm, and 20–25 mm).

### 2.3. CO_2_-Accelerated Carbonation Experiment for RCA Samples

For the CO_2_-accelerated carbonation modification test studying the influence of two important material factors, OCS and CAS, on the physical properties of RCAs, the RCA test specimens, prepared as described above, were selected, and standard accelerated carbonation curing conditions with an ambient temperature of 20 °C, a relative humidity of 70%, and a CO_2_ concentration of 20% were set. Three parallel specimens of RCAs with various properties were weighed, each 500 g, and laid flat in a concrete carbonation box for accelerated carbonation. The specimen numbers and specific test conditions relating to the RCAs are detailed in Table 2.

After starting the carbonation test chamber, the beginning mass of each sample was recorded frequently during the accelerated carbonation test; then, later recording intervals gradually became longer until the quality of the sample remained basically unchanged. Here, the accelerated carbonation reaction was considered complete and the test ended. Immediately after the accelerated carbonation of the RCA specimens, the carbonized recycled coarse aggregate (CRCA) specimens were placed in a drying oven at a temperature of 105 ± 5 °C, dried to constant weight, and the mass of each specimen was recorded. Subsequently, three parallel specimens were mixed homogeneously and then re-divided into three parallel specimens to test the various physical property indexes of the RCAs. Finally, with reference to Yang et al. [38], who tested the alkali aggregate properties of RCAs before and after accelerated carbonation, the accelerated carbonation effect of the RCA and CRCA samples was initially qualitatively determined based on a color rendering reaction by titrating RCA and CRCA samples using an alcoholic phenolphthalein solution indicator with a concentration of 1%, as shown in Figure 2. From the figure, the RCA specimens presented with an obvious purple-red color before carbonation, which indicated that the alkaline aggregate properties of the RCAs were significant. After complete carbonation treatment, the RCA samples were colorless, indicating that the active carbonation of RCA by CO_2_ had successfully changed the RCA specimens from alkaline to neutral; thus, the accelerated carbonation of RCA had a significant effect.

### 2.4. Test for Properties of RCA Samples

Three physical property indicators, apparent density, water absorption, and the water content of the RCAs, were tested for this study. Moreover, mass variation and carbonation rate, which are two important indicators that reflect the carbonation rate and degree relating to RCAs, were also measured. The detailed test procedure is described in the specification “Pebbles and gravel for construction (GB 14685-2011)” [39] and related literature reports by Wu et al. [25], Ju et al. [26], and Yang et al. [38]. The actual tests showed that the apparent density and water absorption of NCA and RCA under different conditions were examined, as detailed in Table 3.

## 3. Results and Analysis

### 3.1. Effect of OCS and CAS on Apparent Density of CRCAs

The measured results of the apparent density of NCAs, RCAs, and CRCAs with various OCSs and CASs are shown in Figure 3. As shown in Figure 3, the apparent density of NCAs is greater than that of RCAs and CRCAs, and the apparent density of CRCAs is overall higher by 58.11 kg/m^3^ than RCAs. The reason for this is that, during the accelerated carbonation process, the carbonation product fills the RCA pores and improves the loose and porous structure within the aggregate, ultimately leading to an increase in the apparent density of RCAs.

In order to quantify the degree of improvement in the apparent density of RCAs via accelerated carbonation, the rate of increase in apparent density Δ*ρ_a_* was defined, as shown in Equation (1):(1)$${\Delta \rho }_{a}=\frac{\rho_{a\mathrm{CRCA}}-\rho_{a\mathrm{RCA}}}{\rho_{a\mathrm{RCA}}}\;\times \;100\%$$
where *ρ_a__CRCA_* indicates the apparent density of CRCAs (kg/m^3^) and *ρ_a__RCA_* denotes the apparent density of RCA (kg/m^3^).

The variation rule of Δ*ρ_a_* in relation to OCSs and CASs is shown in Figure 4. 

(1)Effect of OCS on CRCA apparent density

As shown in Figure 4, under completely dry conditions, the Δ*ρ_a_* of RCAs with CASs of 5–10 mm and 10–20 mm after accelerated carbonation showed a tendency to increase and then decrease with increasing OCS, and the Δ*ρ_a_* of RCAs with an OCS of C40 and CASs of 10–20 mm was the largest after carbonation, 3.95%, demonstrating the most significant improvement. After analysis, this was considered to be due to the fact that the RCAs with low OCSs had a low degree of compactness, a high pore distribution, and a limited amount of carbonizable material, while the RCAs with the highest OCS grade (i.e., C50) had a high degree of compactness, which made it more difficult for CO_2_ to penetrate during the carbonation process. Therefore, RCAs with an OCS of C40 contain richer carbonizable materials (CH and C-S-H) and sufficient carbonation space for the best modification.

In addition, the Δ*ρ_a_* before and after the carbonation of RCAs with a CAS of 20–25 mm increases with the increase in OCS, i.e., at this CAS, RCAs with high OCS are better modified via carbonation, which is due to the thinner, older mortar layer wrapped around the surface layer of RCAs with greater CAS. Furthermore, the densification of the old mortar layer with a high degree of densification is better when fully carbonated.

It was found that the variation rule of Δ*ρ_a_* with OCS after the carbonation of RCAs with higher or lower OCS is actually not uniform; its carbonation process is more obviously limited by CAS. From a comprehensive point of view, the Δ*ρ_a_* of RCAs with an OCS of C40 is the best.

(2)Effect of CAS on CRCA apparent density

As shown in Figure 4, under completely dry conditions, Δ*ρ_a_* showed an increasing trend and then decreased with increasing CAS, and the RCAs with a CAS of 10–20 mm and an OCS of C40 had the largest Δ*ρ_a_* after carbonation, 3.95%, which was the most significant improvement. RCAs with a CAS of 10–20 mm in a completely dry state have the highest Δ*ρ_a_* after carbonation compared to other CASs, which is the most suitable CAS for an RCA in terms of carbonation modification. In summary, CAS is the key factor affecting the physical properties of CRCAs. As far as apparent density is concerned, the CAS of RCAs that is suitable for optimal carbonation modification is 10–20 mm.

### 3.2. Effect of OCSs and CASs on Water Absorption of CRCAs

Figure 5 and Figure 6 show the measured results relating to the water absorption of NCAs, RCAs, and CRCAs with various OCSs and CASs. The water absorption of RCAs and CRCAs is much larger than that of NCAs, and the mean value of the difference between the pre- and post-carbonation water absorption of RCA at various OCSs is 1.15%. The reason for this is that during the CO_2_-accelerated carbonation process, the carbonation product fills the pores of RCAs and improves the loose and porous microstructure within the aggregate, decreasing their water absorption capacity.

In order to quantitatively assess the reduction in the degree of RCA water absorption due to the carbonation treatment, the rate of reduction in water absorption Δ*W_a_* was defined as shown in Equation (2):(2)$${\Delta W}_{a}=\frac{W_{a\mathrm{CRCA}}-W_{a\mathrm{RCA}}}{W_{a\mathrm{RCA}}}\;\times \;100\%$$
where *W_a__CRCA_* is the water absorption rate of CRCAs (%), and *W_a__RCA_* is the water absorption rate of RCAs (%).

The variation rule of Δ*W_a_* with OCS and CAS is shown in Figure 7.

(1)Effect of OCS on CRCA water absorption

As shown in Figure 5, before accelerated carbonation, the water absorption of RCAs across all CASs increased with increasing OCS. However, after accelerated carbonation, the water absorption of RCAs with CASs of 5–10 mm and 10–20 mm decreased with the increase in OCS, and the water absorption of RCAs with a CAS of 20–25 mm showed a tendency to increase first and then decrease with increased OCS.

The quantitative analysis presented in Figure 7 shows that the Δ*W_a_* of CRCAs with CASs of 5–10 mm and 10–20 mm increased with the increase in OCS, and the CRCAs with an OCS of C50 and a CAS of 5–10 mm had the greatest Δ*W_a_*, 29.56%, which is the most significant improvement. The Δ*W_a_* of CRCAs with a CAS of 20–25 mm showed a trend to decrease and then increase with the increase in OCS, and the CRCA with an OCS of C50 had the largest Δ*W_a_*, 23.38%.

In summary, RCAs with a higher OCS (i.e., C50) are carbonized, reducing Δ*W_a_* to a greater extent.

(2)Effect of CAS on CRCA water absorption

From Figure 6, the water absorption of RCA before and after accelerated carbonation decreased with the increase in CAS. The water absorption of CRCAs with an OCS of C30 decreased with the increase in CAS, while the water absorption of CRCAs with OCSs of C40 and C50 showed a decreasing trend and then increased with the increase in CAS. In addition, the CAS corresponding to the maximum value of water absorption of the CRCA was 5–10 mm for all three OCSs.

The quantitative analysis presented in Figure 7 shows that, under completely dry conditions, the Δ*W_a_* of the CRCAs with an OCS of C30 shows a tendency of decreasing and then increasing with the increase in CAS; the Δ*W_a_* of the CRCAs with an OCS of C40 shows a tendency of increasing and then decreasing with increased CAS, and the Δ*W_a_* of the CRCA with an OCS of C50 shows a tendency of decreasing with increasing CAS. The Δ*W_a_* of CRCAs under different OCSs did not show a clear pattern of change in relation to CAS. Taken together, from the magnitude of the value of Δ*W_a_*, CRCAs with relatively smaller CASs (i.e., 5–10 mm and 10–20 mm) showed a greater Δ*W_a_*.

### 3.3. Effect of OCSs and CASs on Moisture Content of CRCAs

Figure 8 shows the measured moisture content of RCAs with different OCSs and CASs before and after accelerated carbonation. From Figure 8, the moisture content of CRCAs increased significantly due to the absorption of water in the carbonation environment.

(1)Effect of OCS on CRCA moisture content

At the same CAS, the water content of CRCAs increased with the increase in OCS and follows a nearly linear trend. This is because the higher the OCS, the more hydration products the RCA contains, and the higher the water uptake capacity during the carbonation process.

(2)Effect of CAS on CRCA moisture content

At the same OCS, the water content of CRCA increases gradually with increasing CAS. This is due to the fact that the larger the CAS, the larger the water content volume of RCAs during carbonation.

From the above analysis of the effects of OCS and CAS on the water content before and after the accelerated carbonation of RCA, the change in the water content of RCAs depends greatly on the relative humidity in the carbonation environment and is not substantially related to OCS and CAS. Therefore, water content should not be directly used as a quantitative index to judge the degree of RCA carbonation modification.

### 3.4. Effect of OCS and CAS on Mass Variation of CRCAs

(1)Effect of OCSs on CRCA mass variation

Figure 9 shows that, in any case, the mass of RCA samples with increasing carbonation time shows the following trend: “rapid increase, followed by a slow increase, and ultimately tends to flatten out”. When the mass of the specimens no longer increases with carbonation time, the carbonation reaction is complete. With lower OCS, the RCA specimens reach their carbonation endpoint faster. With CASs of 5–10 mm and 20–25 mm, the time required for the carbonation of RCA specimens to be complete, as well as their mass when fully carbonized, increase with increasing OCS. When the CAS is 10–20 mm, the time required for the carbonation of RCA corresponding to the OCSs of C40 and C50 to be completed, as well as the mass at full carbonation, are very close to each other, and the mass at full RCA carbonation, corresponding to C30, is lower compared to the C40 and C50 specimens.

Figure 10 demonstrates the variation rule relating to RCAs with various OCSs and CASs in terms of mass increase before and after carbonation. From the figure, the mass increase in RCA after accelerated carbonation increases with increasing OCS at any CAS, and the mass increase in the RCA with an OCS of C50 and a CAS of 5–10 mm is the highest, 22.6 g.

In summary, the higher the OCS, the greater the mass increase in RCA, but the longer it takes to complete carbonation.

(2)Effect of CASs on CRCA mass variation

Overall, as can be seen in Figure 9, when the CAS is smaller, the RCA reaches the carbonation endpoint faster. This is because the smaller the CAS of an RCA, the larger its specific surface area and the larger the contact area with the carbonation mediums of CO_2_ and water; thus, the carbonation products generated are less of a hindrance to the subsequent carbonation reaction. With an OCS of C30, the time required for the carbonation of RCA specimens with CASs of 5–10 mm and 10–20 mm to complete is close to and shorter than that of 20–25 mm; when the OCSs are C40 and C50, the time required for the carbonation of RCA specimens with CASs of 10–20 mm and 20–25 mm to complete is close to and longer than that of 5–10 mm.

From Figure 10, the mass increase in the CRCA with an OCS of C30 increases with the increasing CAS; the mass increase in the CRCA with an OCS of C40 shows a tendency to increase and then decrease with increasing CAS; and the mass increase in the CRCA with an OCS of C50 decreases with increasing CAS. By averaging the mass of RCA gains at 5–10 mm, 10–20 mm, and 20–25 mm corresponding to OCSs of C30, C40, and C50, respectively, the mass of RCA gain values at 5–10 mm, 10–20 mm, and 20–25 mm can be calculated as 16 g, 17.5 g, and 17.47 g, respectively.

Thus, overall, the degree of mass increase in RCA is comparable for CASs of 10–20 mm and 20–25 mm.

### 3.5. Effect of OCS and CAS on Carbonation Ratio of CRCAs

(1)Effect of OCSs on CRCA carbonation ratio

From Figure 11, when the OCS is C40, the carbonation ratio of the RCA is the highest; the magnitude of the values exceeded 20%, and the average value of the carbonation ratio of the specimen reaches 23.59%. Among them, the carbonation ratio of an RCA with an OCS of C40 and a CAS of 10–20 mm was the largest, 28.01%. Although the carbonation ratio of an RCA with an OCS of C50 with all CASs also exceeded 20%; the mean value of the carbonation ratio of the specimens was 23.13%, which was slightly lower than that of the RCA with a strength of C40. The specimen with the lowest carbonation ratio is the RCA with an OCS of C30, which achieved values of less than 20% of the average value of the carbonation ratio, which is only 15.37%.

The analysis provided in this paper suggests that this is a result of the combination of the carbonizable material present in the RCA and the overall degree of densification provided by the different OCSs. Low-OCS (i.e., C30) RCAs are generally loose, with limited hydration products. They have a limited ability to capture and absorb CO_2_ and a low overall carbonation ratio. The hardened cement slurry attached to the high-OCS (i.e., C50) RCA contains more carbonizable substances, but the overall density is relatively high, and the carbonation medium is difficult to permeate, resulting in a more difficult carbonation reaction. Medium-OCS (i.e., C40) RCAs achieve balance between the two, and the ratio of their carbonation is the highest.

(2)Effect of CAS on CRCA carbonation ratio

From Figure 11, with a CAS of 10–20 mm, the carbonation ratio of the RCA is the highest, with values exceeding 15%, and the average value of the carbonation ratio of the specimens reaches 22.09%. The next highest carbonation ratio in terms of RCA is when the CAS is 20–25 mm; its carbonation ratio also exceeds 15%, and the average value of the carbonation ratio is 20.59%. The CAS of the RCA with the lowest carbonation ratio is 5–10 mm, and its carbonation ratios are all below 15%, with an average carbonation ratio of only 19.41%.

Although RCAs with larger CASs have a higher density and are difficult for carbonation media to penetrate, there is also more hardened cement slurry covering the surface, meaning that the material that can be used for carbonation is more abundant. Therefore, RCAs with a medium CAS (i.e., 10–20 mm) can ensure a certain amount of carbonation space and provide more abundant carbonation material, which has the best modification effect.

### 3.6. The Best Property Enhancements of CRCAs

By comprehensively analyzing the variation rules of each physical property index (apparent density, water absorption, mass change, and carbonation ratio) of the RCAs before and after carbonation in Section 3.1, Section 3.2, Section 3.3, Section 3.4 and Section 3.5, the CASs corresponding to the RCAs with different OCSs in the case of optimal carbonation modification effects were summarized and are shown in Table 4.

As shown in Table 4, the corresponding relationships between OCS and CAS with optimal carbonation modification of RCA are as follows: C30 and 20–25 mm, C40 and 10–20 mm, and C50 and 5–10 mm. Subsequently, comparative graphs were plotted to obtain Δ*ρ_a_*, Δ*W_a_*, and the carbonation ratio of the RCA specimens for the above three combinations, as shown in Figure 12. From the figure, the C40 and 10–20 mm RCA samples showed the best improvement in all of the performance indicators when compared to the combination of C30 and 20–25 mm and C50 and 5–10 mm.

### 3.7. Microstructure (SEM Analysis) of the RCAs before and after Carbonation

For SEM analysis, the RCA samples were dried in a vacuum environment for at least 3 days. Subsequently, the samples were coated with gold [40] before the SEM test. In this paper, the SEM analysis of the RCA samples was performed with a ZEISS Sigma 300 (Oberkochen, Germany) at a working voltage of 15 kV. Figure 13 shows the SEM images of the microstructures of the RCA samples, i.e., C30 and 20–25 mm, C40 and 10–20 mm, and C50 and 5–10 mm, before and after CO_2_-accelerated carbonation.

The pores and microcracks within the OM and OITZ inside the CRCAs are filled with densification carbonation products, including CaCO_3_ and silica gel, and the microstructures of the OM and OITZ of CRCA are denser than those of the RCA, as shown in Figure 13b,d,f. Therefore, the macroscopic properties, such as mass increase (Δm), apparent density (*ρ_a_*), and water absorption (*W_a_*), for the CRCAs are better than those of the RCAs before ACE. In addition, from the SEM images, the microstructural densification degrees for the OM and OITZ of C40 and 10–20 mm CRCAs are greater than those of the C50 and 5–10 mm CRCAs as well as the C30 and 20–25 mm CRCAs, which reasonably clarifies the objective reason for the enhancement of the macroscopic properties of C40 and 10–20 mm CRCAs and shows the essential mechanism that occurs at the material microstructural level.

## 4. Conclusions

In this study, CO_2_-accelerated carbonation modifications of RCAs under completely dry conditions with various OCSs and CASs were carried out. The effects of OCS and CAS on the physical property indexes of CRCAs, such as apparent density, water absorption, water content, mass increase, and carbonation ratio, were revealed, and the OCS and CAS results corresponding to the best accelerated carbonation effects of RCAs were determined. Some significant conclusions are as follows:(1)The CRCAs obtained via the CO_2_-accelerated carbonization of RCAs with various OCSs and CASs were improved to varying degrees in terms of apparent density, water absorption, quality change, and carbonization ratio. This indicates that the CO_2_-based accelerated carbonization method can improve the performance of RCAs.(2)Based on the degree of improvement in the performance indexes, such as apparent density, water absorption, mass change, and carbonation ratio, before and after the accelerated carbonation of RCA specimens, the OCSs and CASs with the optimal effect of RCA carbonation modification were clarified as follows: C30 and 20–25 mm, C40 and 10–20 mm, and C50 and 5–10 mm, respectively.(3)Through the comparative analysis of the macroscopic properties and microscopic morphology of RCAs with optimal carbonation modification effects, it was finally determined that, when the OCS is C40 and the CAS is 10–20 mm, the performance improvement of RCAs is higher than that of C30 and 20–25 mm and C50 and 5–10 mm.

## Figures and Tables

**Figure 1 materials-17-03567-f001:**
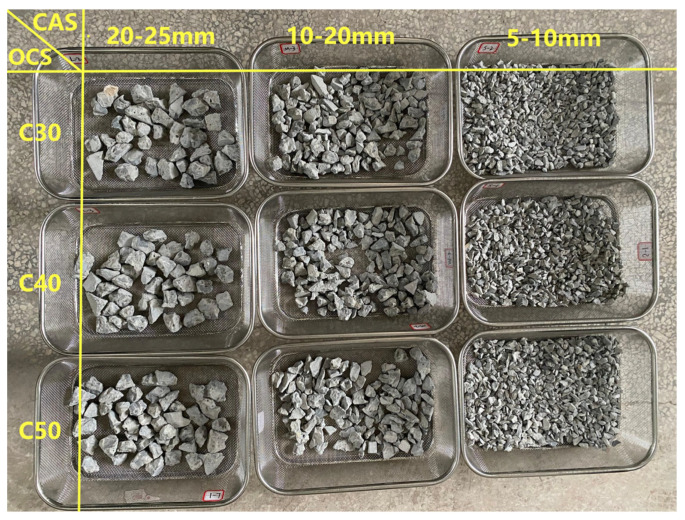
RCAs with different OCSs (C30, C40, and C50) and various CASs (20–25 mm, 10–20 mm, and 5–10 mm).

**Figure 2 materials-17-03567-f002:**
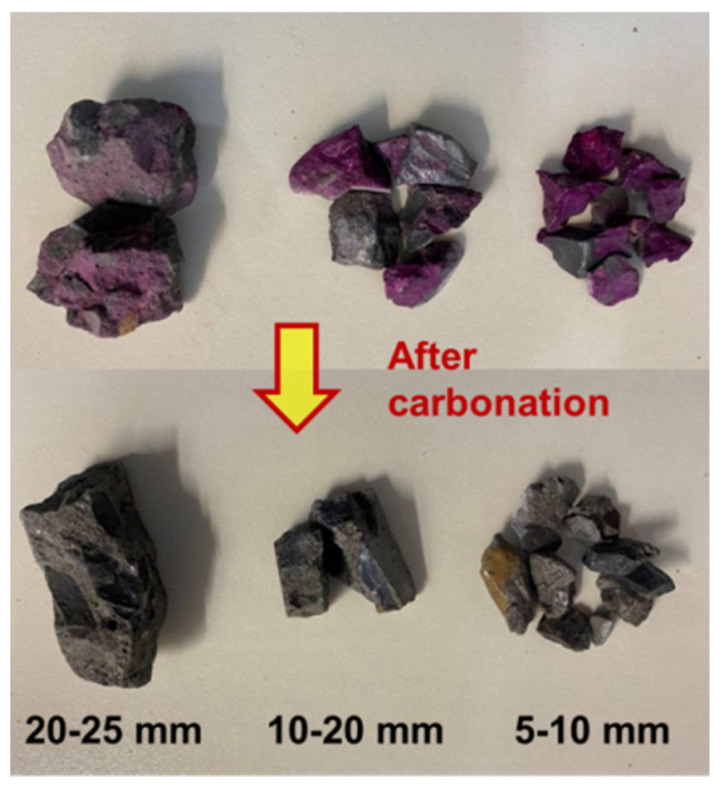
Chromogenic reaction for the RCA samples with CASs of 5–10 mm, 10–20 mm, and 20–25 mm before and after ACE.

**Figure 3 materials-17-03567-f003:**
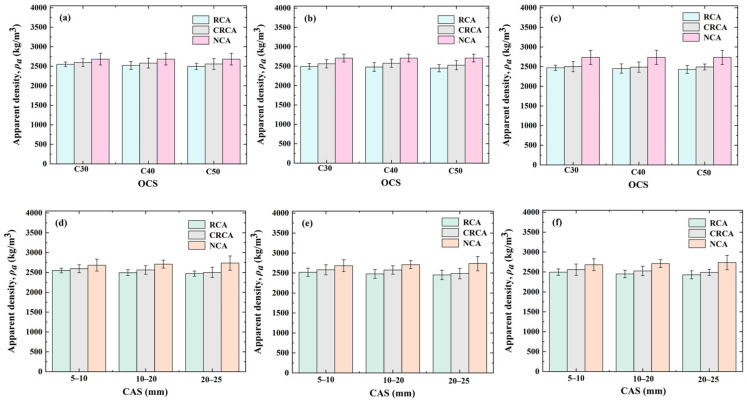
Measured values of NCA, RCA, and CRCA apparent density for various OCSs and CASs: (**a**) 5–10 mm, (**b**) 10–20 mm, (**c**) 20–25 mm, (**d**) C30, (**e**) C40, and (**f**) C50.

**Figure 4 materials-17-03567-f004:**
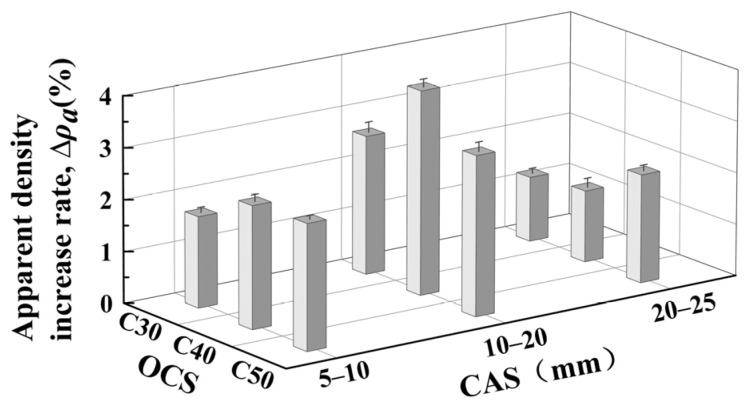
Δ*ρ_a_* before and after accelerated carbonation of RCAs for various OCSs and CASs.

**Figure 5 materials-17-03567-f005:**
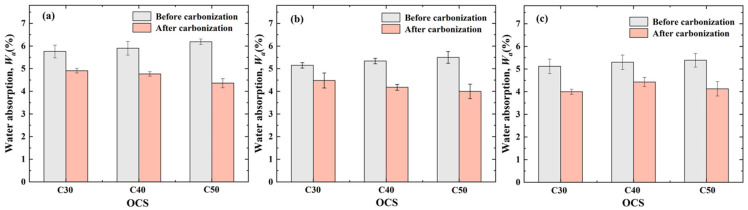
Measured values of water absorption of RCA and CRCA for various CASs: (**a**) 5–10 mm; (**b**) 10–20 mm; (**c**) 20–25 mm.

**Figure 6 materials-17-03567-f006:**
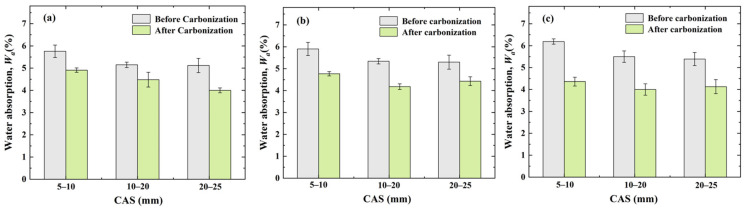
Measured values of water absorption of RCA and CRCA for various OCSs: (**a**) C30; (**b**) C40; (**c**) C50.

**Figure 7 materials-17-03567-f007:**
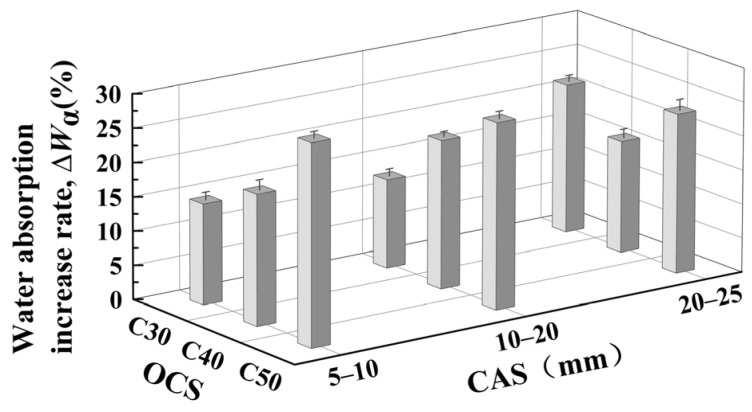
Δ*W_a_* before and after accelerated carbonation of RCA with different OCSs and CASs.

**Figure 8 materials-17-03567-f008:**
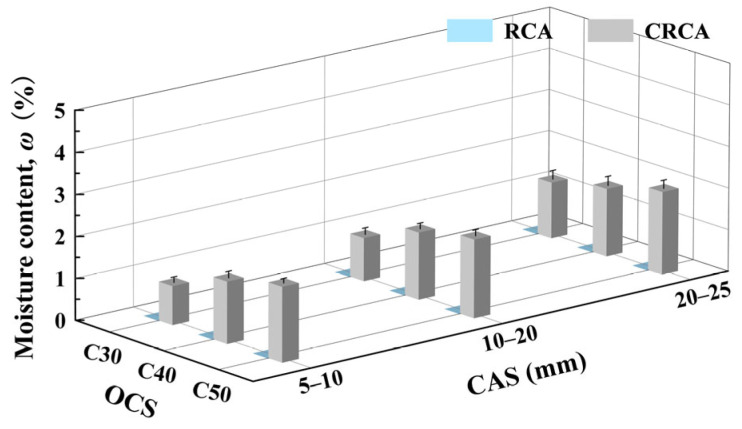
Measured values of water content of RCA and CRCA for different OCSs and CASs.

**Figure 9 materials-17-03567-f009:**
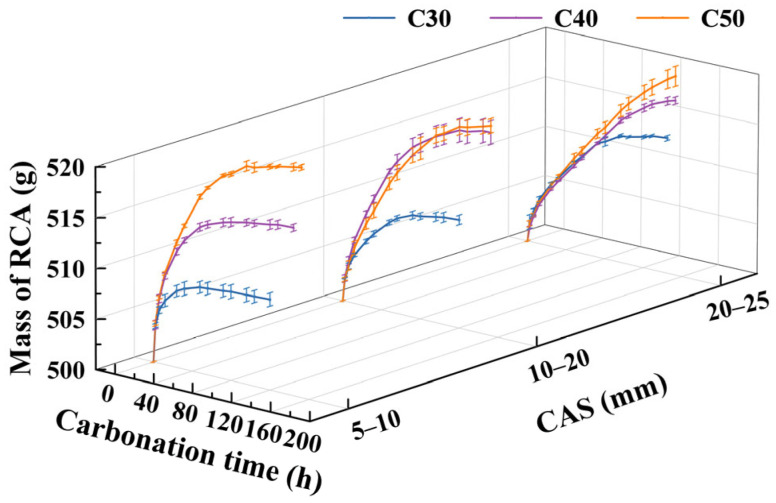
Time-varying pattern of RCA quality for various OCSs and CASs.

**Figure 10 materials-17-03567-f010:**
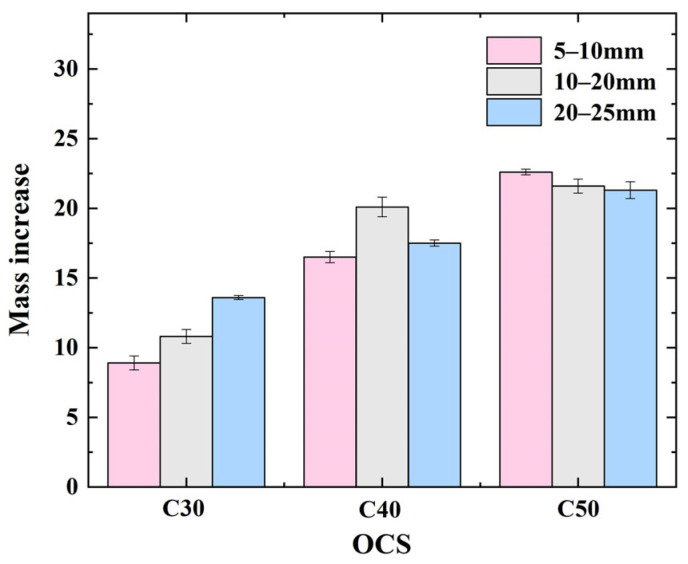
Mass increase in RCA before and after carbonation.

**Figure 11 materials-17-03567-f011:**
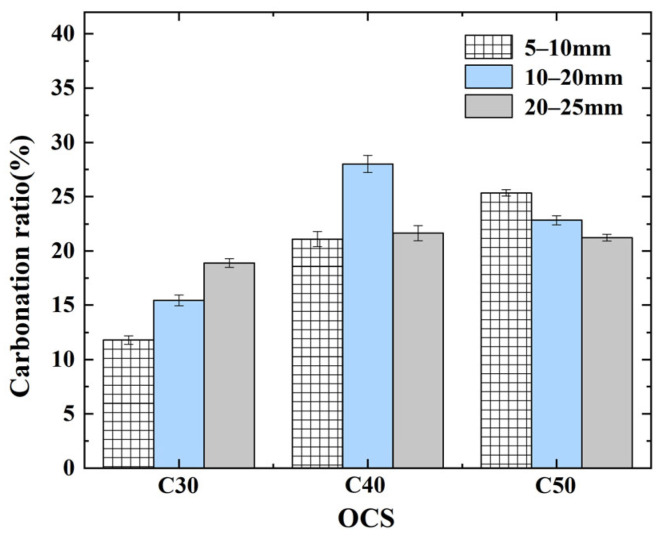
Carbonation ratio of RCAs.

**Figure 12 materials-17-03567-f012:**
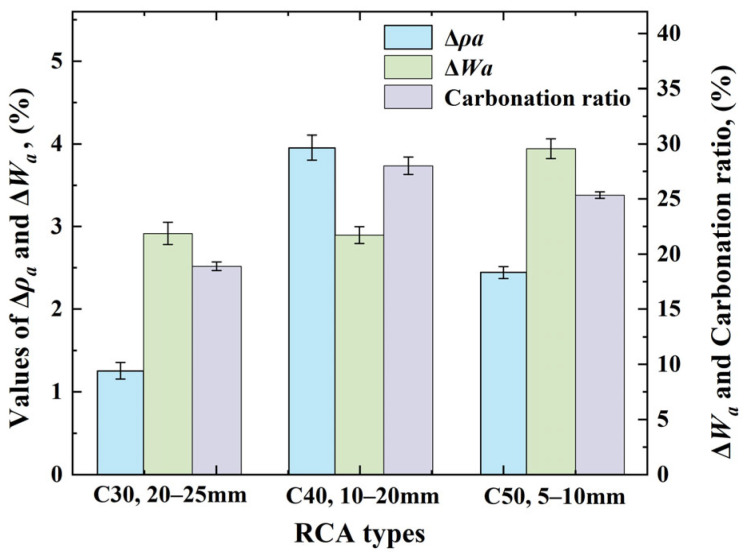
Types of RCAs with optimal carbonation modifications and their performance improvements.

**Figure 13 materials-17-03567-f013:**
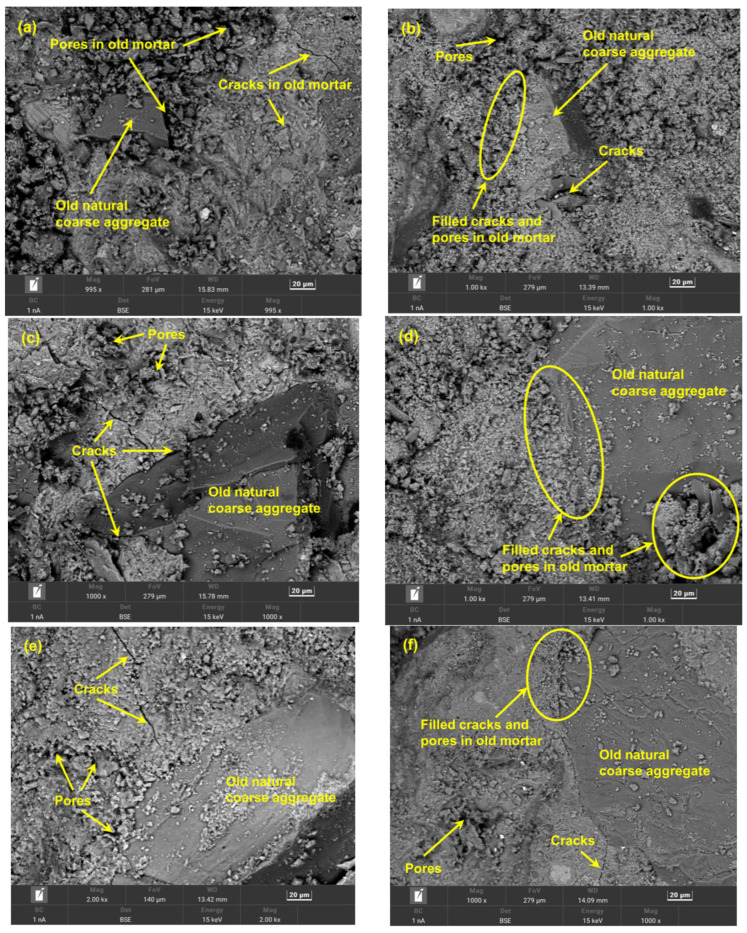
SEM images for the RCA samples before and after ACE. (**a**) RCA with C50 and 5–10 mm; (**b**) CRCA with C50 and 5–10 mm; (**c**) RCA with C40 and 10–20 mm; (**d**) CRCA with C40 and 10–20 mm; (**e**) RCA with C30 and 20–25 mm; (**f**) CRCA with C30 and 20–25 mm.

**Table 1 materials-17-03567-t001:** Mix proportions of prepared raw RCA concrete.

Strength Grade	*w*/*c*	Components of Raw Materials (kg/m^3^)			
		Water	Cement	Coarse Aggregate	Fine Aggregate
C50	0.4	195	488	1134	584
C40	0.5	195	390	1198	617
C30	0.6	195	325	1241	639

**Table 2 materials-17-03567-t002:** Specimen properties of RCAs subjected to CO_2_ carbonation modification.

Specimen Properties	Original Concrete Strength, OCS (MPa)	Coarse Aggregate Size, CAS (mm)
SP-30-10	C30	5–10
SP-30-20		10–20
SP-30-25		20–25
SP-40-10	C40	5–10
SP-40-20		10–20
SP-40-25		20–25
SP-50-10	C50	5–10
SP-50-20		10–20
SP-50-25		20–25

**Table 3 materials-17-03567-t003:** Apparent density and water absorption for NCA and RCA with different OCSs and CASs.

Types of CA	OCS (MPa)	CAS (mm)	*ρ_a_* (kg/m^3^)	*W_a_* (%)
NCA	—	5–10	2682	0.34
		10–20	2710	0.33
		20–25	2736	0.31
RCA	C30	5–10	2549	5.76
		10–20	2495	5.15
		20–25	2470	5.12
	C40	5–10	2521	5.90
		10–20	2479	5.34
		20–25	2453	5.30
	C50	5–10	2497	6.19
		10–20	2450	5.50
		20–25	2430	5.39

**Table 4 materials-17-03567-t004:** OCS and CAS corresponding to the optimal RCA carbonation modification effect.

OCS	Δ*ρ_a_* (%)	Δ*W_a_* (%)	Mass Variation (g)	Carbonation Ratio (%)
C30	10–20 mm	20–25 mm	20–25 mm	20–25 mm
C40	10–20 mm	10–20 mm	10–20 mm	10–20 mm
C50	10–20 mm	5–10 mm	5–10 mm	5–10 mm

## Data Availability

The original contributions presented in the study are included in the article, further inquiries can be directed to the corresponding authors.

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
