# Peer review of "CO2-Accelerated Carbonation Modification for Recycled Coarse Aggregate with Various Original Concrete Strengths and Coarse Aggregate Sizes"

_materials, 2024, doi:10.3390/ma17143567_

Round 1

Reviewer 1 Report

Comments and Suggestions for Authors

In general, the topic of the paper is of interest both for scientific community and designers also considering the importance of the sustainability in the new construction. Furthermore, the manuscript appears well-organized in its different Sections and the laboratory tests and the related results obtained are clearly described and discussed in the text. For these reasons, it is opinion of this reviewer that the manuscript can be considered for publication in Materials after the following minor corrections/improvements:

- in the introduction consider as reported in 10.1007/978-3-031-37123-3_21,
 10.1016/j.conbuildmat.2013.04.051

- How the authors have considered the problem of the recycled aggregates water absorption?

- How the w/c ratio of the considered mixes design influences the mechanical properties of the concrete?

- From the practical point of view, the workability and the durability of the considered mixes design differ from those of a standard concrete mix design?

- in the conclusions better highlights the original aspects of the research work and of the obtained results 

Comments on the Quality of English Language

The paper appears well-written 

Author Response

We would like to appreciate your insightful and constructive comments on our manuscript. We have fully revised and improved the contents of our paper according to your suggestions. The explanations for your mentioned questions in our manuscript are as follows:

In general, the topic of the paper is of interest both for scientific community and designers also considering the importance of the sustainability in the new construction. Furthermore, the manuscript appears well-organized in its different Sections and the laboratory tests and the related results obtained are clearly described and discussed in the text. For these reasons, it is opinion of this reviewer that the manuscript can be considered for publication in Materials after the following minor corrections/improvements:

Point 1: In the introduction consider as reported in 10.1007/978-3-031-37123-3_21, 10.1016/j.conbuildmat.2013.04.051

Response 1: Esteemed reviewer, thank you very much for your valuable suggestions. We have cited the two references you suggested in our revised manuscript, please refer to the references 2 and 9 in Section "1.Introduction" of the revised manuscript. Thank you very much for your constructive and insightful suggestions.

Point 2: How the authors have considered the problem of the recycled aggregates water absorption?

Response 2: Esteemed reviewer, RCA itself is highly absorbent and this has been reported in the open literature. Existing studies (Wu et al., 2023; Ju et al., 2024) show that , RCA in the completely dry state has the strongest water absorption and can quickly absorb water and CO2 in the environment during the accelerated carbonation process, making the carbonation modification effect of RCA in this state the best. Therefore, RCA with different original concrete strength and coarse aggregate size under completely dry condition is selected as the research object in this paper to explore the influence of CO2-based accelerated carbonation method on the carbonation modification degree of the aforementioned RCA samples. For details, please refer to Section 2.2 of the revised manuscript. All the authors of this paper sincerely entreat that you can accept our consideration for this question, thank you very much.

[1] Wu, L.; Zhang, W.; Jiang, H.; Ju, X.; Guan, L.; Liu, H.; Chen, S. Synergistic Effects of Environmental Relative Humidity and Initial Water Content of Recycled Concrete Aggregate on the Improvement in Properties via Carbonation Reactions. Materials 2023, 16, 5251. [CrossRef]

[2] Ju, X.; Wu, L.; Liu, M.; Jiang, H.; Zhang, W.; Guan, L.; Chen, X.; Fan, X. Influence of the Original Concrete Strength and Initial Moisture Condition on the Properties Improvement of Recycled Coarse Aggregate via Accelerated Carbonation Reactions. Materials 2024, 17, 706. [CrossRef]

Point 3: How the w/c ratio of the considered mixes design influences the mechanical properties of the concrete?

Response 3: Esteemed reviewer, generally speaking, concrete w/c will significantly affect its mechanical properties, the lower the w/c, the higher the strength of the concrete, the better the mechanical properties. For this paper’s investigations, the accelerated carbonation modification effect of RCA prepared with different original concrete strengths has been studied, and the specific influence degrees are detailed in Chapter 3 and conclusions of this paper. All the authors of this paper sincerely entreat that you can accept our consideration for this question, thank you very much.

Point 4: From the practical point of view, the workability and the durability of the considered mixes design differ from those of a standard concrete mix design?

Response 4: Esteemed reviewer, the concrete mix used in this paper is designed in full accordance with the Chinese standards, and the workability and durability of concrete are different from the concrete mixture used in practice more or less. However, considering that scientific research focuses on exploring the objective intrinsic law of RCA accelerated carbonation modification, the raw materials we use for experimental studies have to use the standard concrete designed in accordance with the standards. In response to your valuable suggestions, we will select actual concrete raw materials in the future research process to carry out more in-depth exploration. All the authors of this paper sincerely entreat that you can accept our consideration for this question, thank you very much.

Point 5: In the conclusions better highlights the original aspects of the research work and of the obtained results

Response 5: Esteemed reviewer, thank you very much for your feedback to us. We have revised the content of the conclusion to better highlight the original aspects of the research work and the results obtained in the conclusion. Moreover, we have already submitted this revised manuscript to a professional language editing service company (Multidisciplinary Digital Publishing Institute) to help us improve the language of the manuscript. This may certify that this revised manuscript was edited for proper English language, grammar, punctuation, spelling, and overall style by one or more of the highly qualified native English speaking editors at “Multidisciplinary Digital Publishing Institute”. We believe that the language of the edited manuscript should be English-ready for publication. All the authors of this paper sincerely entreat that you can accept our consideration for this question, thank you very much.

What is said above are the entire responses to all your questions, and we greatly appreciate your thorough review and hope that this revised new manuscript in its present form will be acceptable for publication at Materials, thank you and best wishes.

Sincerely yours,

Wei Qin and Xinhui Fan on behalf of all the authors

Reviewer 2 Report

Comments and Suggestions for Authors

CO2-accelerated carbonation modification for recycled coarse aggregate with various original concrete strengths and aggregate particle diameters

materials-3087912

The manuscript presents an experimental study of the material properties of recycled concrete aggregates and the effects of accelerated carbonation. The experimental campaign included investigations of densities, water absorption and microstructures. The results showed that carbonation has several positive effects on the material properties of recycled concrete aggregates.

In the abstract there is a very long and complicated sentence, using the abbreviations OCS, RCA and APD four times, which should be rephrased for clarity: “For this purpose, accelerated carbonation tests were conducted on recycled coarse aggregate (RCA) samples with various OCSs and APDs in the initial state of complete drying, revealing the changing rules of the physical property indexes of RCA with the OCS and APD before and after the accelerated carbonation, clarifying the degree of influence of OCS and APD on accelerated carbonation modification of RCA, probing the OCS and APD of RCA when accelerated carbonation modification is optimal.”

In the introduction, the abbreviations need to be defined at their first appearance (even though you explain them in the abstract).

Some of the sentences in the introduction keeps repeating the same abbreviations several times, making it quite tough for the readers to read and understand the meaning. For this reason, some of the text in the introduction needs to be rephrased, avoiding unnecessary repeating of the same abbreviations.  

On line 37-39 I think you mean NCA instead of RA, otherwise your statement doesn’t make sense.

After reference 6 I propose the following addition: “Sustainability and circularity concepts have become key factors for developing the concrete industry and incorporating more recycled materials and by-products is considered as the way forward [x]” [x] https://doi.org/10.1016/j.dibe.2023.100177  

In the materials and experiments section, some of the superscripts are not correct, for example for cubic meters.

The term aggregate particle diameter (APD) is not very common and I would prefer to call it aggregate size.

Regarding the production of the recycled aggregates, it is not clear how and for how long the original concrete was cured before it was crushed. If the compressive strength was tested before the crushing, it should be presented in the paper.

The NCA presented in Table 3 are not explained in the paper. In section 2.1 you only define fine aggregates and coarse aggregates (up to 20 mm), but in Table 3 there are three grades of coarse aggregates (up to 25 mm). This needs clarification.

Is it possible to merge Figure 6 a-c into one Figure with all results, and the same for Figure 7 a-c?

The first sentence of the conclusions needs to be rephrased for improved clarity.

The information regarding appendix A and B is unnecessary for the paper and should be removed.

The topic of the paper is very interesting and suitable for publishing in Materials. By incorporating the suggestions above, the paper will become a good contribution to the journal.

Comments on the Quality of English Language

There are several long and complicated sentences, as well as unnecessary repetition of abbreviations. 

Author Response

We would like to appreciate your insightful and constructive comments on our manuscript. We have fully revised and improved the contents of our paper according to your suggestions. The explanations for your mentioned questions in our manuscript are as follows:

The manuscript presents an experimental study of the material properties of recycled concrete aggregates and the effects of accelerated carbonation. The experimental campaign included investigations of densities, water absorption and microstructures. The results showed that carbonation has several positive effects on the material properties of recycled concrete aggregates.

Point 1: In the abstract there is a very long and complicated sentence, using the abbreviations OCS, RCA and APD four times, which should be rephrased for clarity: “For this purpose, accelerated carbonation tests were conducted on recycled coarse aggregate (RCA) samples with various OCSs and APDs in the initial state of complete drying, revealing the changing rules of the physical property indexes of RCA with the OCS and APD before and after the accelerated carbonation, clarifying the degree of influence of OCS and APD on accelerated carbonation modification of RCA, probing the OCS and APD of RCA when accelerated carbonation modification is optimal.”

Response 1: Esteemed reviewer, we have revised this paragraph to “For this purpose, accelerated carbonation tests on recycled coarse aggregate (RCA) samples under completely dry conditions were carried out, and the variation laws for the physical property indicators of RCA samples before and after accelerated carbonation versus the OCS and CAS were revealed. Moreover, the influence degrees of the two factors, OCS and CAS, on the property enhancement of RCAs after accelerated carbonation were clarified, and the results of OCS and CAS corresponding to the best accelerated carbonation effects of RCAs were determined.”, and the aforementioned variations have been highlighted in yellow background in the “Abstract” of our revised manuscript. Thank you very much for your constructive and insightful suggestions.

Point 2: In the introduction, the abbreviations need to be defined at their first appearance (even though you explain them in the abstract).

Response 2: Esteemed reviewer, we feel really sorry for this mistake, and we have made necessary changes to this issue in our revised manuscript. Thank you very much for your constructive and insightful suggestions.

Point 3: Some of the sentences in the introduction keeps repeating the same abbreviations several times, making it quite tough for the readers to read and understand the meaning. For this reason, some of the text in the introduction needs to be rephrased, avoiding unnecessary repeating of the same abbreviations.  

Response 3: Esteemed reviewer, we feel really sorry for these English language questions, and we have already submitted this revised manuscript to a professional language editing service company (Multidisciplinary Digital Publishing Institute) to help us improve the language of the manuscript. An “Editorial Certificate” was received from “Multidisciplinary Digital Publishing Institute”. This “Editorial Certificate” may certify that this revised manuscript was edited for proper English language, grammar, punctuation, spelling, and overall style by one or more of the highly qualified native English speaking editors at “Multidisciplinary Digital Publishing Institute”. We believe that the language of the edited manuscript should be English-ready for publication.

All the authors of this paper sincerely entreat that you can accept our consideration for this question, thank you very much.

Point 4: On line 37-39 I think you mean NCA instead of RA, otherwise your statement doesn’t make sense.

Response 4: Esteemed reviewer, we regret this error and we have made the necessary changes to the issue and reviewed the full text to ensure that this error does not occur elsewhere. For the specific changes, please see the first paragraph of the introduction to this paper highlighted in yellow background. Thank you very much.

Point 5: After reference 6 I propose the following addition: ”Sustainability and circularity concepts have become key factors for developing the concrete industry and incorporating more recycled materials and by-products is considered as the way forward [x]” [x] https://doi.org/10.1016/j.dibe.2023.100177

Response 5: Esteemed reviewer, your suggestion has been a great help to improve our manuscript. We have already added the related contents in Section “1. Introduction” of our revised manuscript. For the specific changes, please see the first paragraph of the introduction and reference 8 where it is highlighted in yellow background. Thank you very much for your constructive and insightful suggestions. 

Point 6: In the materials and experiments section, some of the superscripts are not correct, for example for cubic meters.

Response 6: Esteemed reviewer, we regret this error. We have made the necessary changes to the issue and reviewed the full text to ensure that this error does not occur elsewhere. Thank you very much.

Point 7: The term aggregate particle diameter (APD) is not very common and I would prefer to call it aggregate size.

Response 7: Esteemed reviewer, we have already revised the "aggregate particle diameter (APD)" and used to replace the "coarse aggregate size (CAZ)" in our revised manuscript. Thank you very much for your constructive and insightful suggestions.

Point 8: Regarding the production of the recycled aggregates, it is not clear how and for how long the original concrete was cured before it was crushed. If the compressive strength was tested before the crushing, it should be presented in the paper.

Response 8: Esteemed reviewer, we have provided the additional contents on the original concrete specimens curing to the preparation of RCA specimens and the compressive strength test results of concrete specimens. For details, see the second paragraph of Section 2.1 of our revised manuscript (highlighted in yellow background). Thank you very much for your constructive and insightful suggestions.

Point 9: The NCA presented in Table 3 are not explained in the paper. In section 2.1 you only define fine aggregates and coarse aggregates (up to 20 mm), but in Table 3 there are three grades of coarse aggregates (up to 25 mm). This needs clarification.

Response 9: Esteemed reviewer, the abbreviation for NCA in Table 3 of our manuscript has also been explained in Section “1 introduction”. In addition, we feel very sorry for the confusion of RCA particle size before and after. The maximum particle size of the coarse aggregate sample used in this study is 25mm. The aforementioned related contents have been modified in our revised manuscript. Thank you very much for your constructive and insightful suggestions.

Point 10: Is it possible to merge Figure 6 a-c into one Figure with all results, and the same for Figure 7 a-c?

Response 10: Esteemed reviewer, during the process of writing this paper, we have ever tried to combine the three figures in Figure 6a-c into one figure as follows. However, because more obvious trends could not be obtained from the 3D figure, it was difficult to intuitively reveal the change rules of the property indicators before and after the acceleration of RCA carbonation, so the 2D figure presented in this paper was finally used for presentation. All the authors of this paper sincerely entreat that you can accept our consideration for this question, thank you very much.

The 3D figure we used earlier looks like this:

Point 11: The first sentence of the conclusions needs to be rephrased for improved clarity.

Response 11: Esteemed reviewer, we have revised the first paragraph of the conclusion to "In this study, CO2-accelerated carbonation modifications of RCAs under completely dry conditions and various OCSs and CASs were carried out. The effects of OCS and CAS on the physical property indexes of CRCAs, such as apparent density, water absorption, water content, mass increase, and carbonation ratio, were revealed, and the OCS and CAS results corresponding to the best accelerated carbonation effects of RCAs were determined. Some significant conclusions are as follows:"... Moreover, we have already submitted this revised manuscript to a professional language editing service company (Multidisciplinary Digital Publishing Institute) to help us improve the language of the manuscript. This may certify that this revised manuscript was edited for proper English language, grammar, punctuation, spelling, and overall style by one or more of the highly qualified native English speaking editors at “Multidisciplinary Digital Publishing Institute”. We believe that the language of the edited manuscript should be English-ready for publication. All the authors of this paper sincerely entreat that you can accept our consideration for this question, thank you very much.

Point 12: The information regarding appendix A and B is unnecessary for the paper and should be removed.

Response 12: Esteemed reviewer, we have already deleted the information in Appendices A and B in the revised manuscript. Thank you very much for your constructive and insightful suggestions.

Point 13: The topic of the paper is very interesting and suitable for publishing in Materials. By incorporating the suggestions above, the paper will become a good contribution to the journal.

Response 13: Esteemed reviewer, thank you very much for your acceptance of our research works, thanks a lot and best wishes. 

What is said above are the entire responses to all your questions, and we greatly appreciate your thorough review and hope that this revised new manuscript in its present form will be acceptable for publication at Materials, thank you and best wishes.

Sincerely yours,

Wei Qin and Xinhui Fan on behalf of all the authors

Reviewer 3 Report

Comments and Suggestions for Authors

Title:

OK

Abstract:

Line 14: It is mentioned that the RAs have a lack of properties, the materials do not have a lack of properties, rather they have inadequate properties for certain uses, or the properties observed are low, minor, etc., correct.

Keywords:

OK

Introduction:

Line 38: grammatical error: sand and gravel, must be spoken in the plural: use amounts

Line 44: plural: "those" instead of "that"

Line 48: several properties are spoken of – use plural.

Line 73: the adjective "larger" is used, which means "larger".

Lines 73 and 74: correct wording.

Line 90: it was meant: completely dry? ( completely dry)

Line 101: the subject of which we are talking (it) is missing

Lines 106-108: the introduction must still be discussed, what is mentioned must be covered in the results and conclusions.

Materials and experiments

Lines 123, 125 and 128: use the correct nomenclature of the units of measurement.

Line 126: mention the reference appropriately.

Line 138: Was it sealed with bags or in bags?

Line 140: Was it designed as three aggregates or with three aggregates?

Line 146: Ditto line 90.

Line 156: write the relative humidity correctly without repetition, use appropriate text and/or nomenclature.

Line 172: If talking about accelerated carbonation, a singular verb should be used: "was" instead of "were"

Lines 185 and 187: three or two are mentioned, correct wording.

Results and analysis

Redundancy and repetition error is repeated in the text: (when using the concepts increasing, toilet absorption, carbonation ratio) improve wording.

Line 249: pore densification is mentioned, by definition these are the presence of voids in the matrix, so we would speak of pore filling, mass increase, pore dimension reduction?, it is suggested to use a more appropriate technical term that describes the observed phenomenon.

The figures must show the error or standard deviation, a statistical analysis of the results is suggested.

Lines 325, 326, 333, 334: improve wording.

The SEM methodology or the parameters used to make the observations shown are not mentioned, it is necessary to describe it in detail to validate and be able to replicate these results, in the same way the use of an image analysis program that corroborates the results shown by this method is requested.

Conclusions: ok

Bibliography: ok

See previous comments.

Comments on the Quality of English Language

There are important errors in the writing. See comments to the authors. A review by an English language specialist is necessary.

Author Response

We would like to appreciate your insightful and constructive comments on our manuscript. We have fully revised and improved the contents of our paper according to your suggestions. The explanations for your mentioned questions in our manuscript are as follows: 

Title: OK

Abstract: 

Line 14: It is mentioned that the RAs have a lack of properties, the materials do not have a lack of properties, rather they have inadequate properties for certain uses, or the properties observed are low, minor, etc., correct.

Response: Esteemed reviewer, we have revised the content of the revised abstract as follows: “Due to the fact that the physical and mechanical properties of waste concrete made of recycled aggregates (RAs) differ greatly, it is difficult to use directly as a raw material for forced reinforced concrete (RC) components, which greatly restricts the popularization and application of RAs in actual projects.” (Please see the abstract highlighted in green.) All the authors of this paper sincerely entreat that you can accept our consideration for this question, thank you very much.

Keywords: OK

Introduction:

Line 38: grammatical error: sand and gravel, must be spoken in the plural: use amounts

Response: Esteemed reviewer, we have corrected the error to "large amounts of" in the corresponding positions of our revised manuscript. Thank you very much for your constructive and insightful suggestions.

Line 44: plural: "those" instead of "that"

Response: Esteemed reviewer, we have already corrected the error as "those" in the corresponding position of our revised manuscript. Thank you very much for your constructive and insightful suggestions.

Line 48: several properties are spoken of – use plural.

Response: Esteemed reviewer, we have corrected the error to "properties" at the correct location for our revised manuscript. Thank you very much for your constructive and insightful suggestions.

Line 73: the adjective "larger" is used, which means "larger".

Response: Esteemed reviewer, we have corrected the error to "large" in the corresponding position of our revised manuscript. Thank you very much for your constructive and insightful suggestions.

Lines 73 and 74: correct wording.

Response: Esteemed reviewer, we feel really sorry for these English language questions, and we have already submitted this revised manuscript to a professional language editing service company (Multidisciplinary Digital Publishing Institute) to help us improve the language of the manuscript. This may certify that this revised manuscript was edited for proper English language, grammar, punctuation, spelling, and overall style by one or more of the highly qualified native English speaking editors at “Multidisciplinary Digital Publishing Institute”. We believe that the language of the edited manuscript should be English-ready for publication. All the authors of this paper sincerely entreat that you can accept our consideration for this question, thank you very much.

Line 90: it was meant: completely dry? ( completely dry)

Response: Esteemed reviewer, we have corrected the error to "completely dry" in the corresponding position of our revised manuscript. Thank you very much for your constructive and insightful suggestions.

Line 101: the subject of which we are talking (it) is missing

Response: Esteemed reviewer, we have modified the corresponding location in this paper, the revised content in “By carrying out an accelerated carbonation tests on the RCA samples with a dry IMC, the change laws of the physical property indexes of the RCAs before and after accelerated carbonation with OCSs and CASs are revealed, and the extent to which OCSs and CASs affect the accelerated carbonation modification of RCAs are clarified, as well as the OCS and CAS results of RCA corresponding to the best accelerated carbonation modification effects, are determined." (Please see the 5-th paragraph of the introduction, highlighted in green.) Thank you very much for your constructive and insightful suggestions.

Lines 106-108: the introduction must still be discussed, what is mentioned must be covered in the results and conclusions.

Response: Esteemed reviewer, we have refined this section in our revised manuscript and deleted some sentences appropriately to make the introduction, results, and conclusions consistent. All the authors of this paper sincerely entreat that you can accept our consideration for this question, thank you very much.

Materials and experiments

Lines 123, 125 and 128: use the correct nomenclature of the units of measurement.

Response: Esteemed reviewer, we apologize for this series of misrepresentations and have corrected this in our revised manuscript. Thank you very much for your constructive and insightful suggestions.

Line 126: mention the reference appropriately.

Response: Esteemed reviewer, we have specified the specification number of reference [40] in our revised manuscript (Please see second paragraph of section 2.1, highlighted in green). Thank you very much for your constructive and insightful suggestions.

Line 138: Was it sealed with bags or in bags?

Response: Esteemed reviewer, we have changed this part of the content to "The original concrete specimens were crushed separately in the order of various OCSs of C30, C40, C50, and they were promptly put in the preparation bags and sealed for subsequent sieving process." (Please see first paragraph of section 2.2, highlighted in green). Thank you very much.

Line 140: Was it designed as three aggregates or with three aggregates?

Response: Esteemed reviewer, we have changed this part of the content to "The RCA was sieved with three different CASs of 5-10 mm, 10-20 mm, and 20-25 mm, respectively." (Please see second paragraph of section 2.2, highlighted in green). The RCA was designed with three different CAZs, of 5-10 mm, 10-20 mm, 20-25 mm, respectively.

Line 146: Ditto line 90.

Response: Esteemed reviewer, we have corrected the error to "completely dry" at the corresponding place in our revised manuscript. Thank you very much for your constructive and insightful suggestions.

Line 156: write the relative humidity correctly without repetition, use appropriate text and/or nomenclature.

Response: Esteemed reviewer, we have revised it to "a relative humidity of 70%" in the corresponding position of our revised manuscript.Thank you very much for your constructive and insightful suggestions.

Line 172: If talking about accelerated carbonation, a singular verb should be used: "was" instead of "were"

Response: Esteemed reviewer, we have revised it to "was” at the corresponding position in our revised manuscript. Thank you very much for your constructive and insightful suggestions.

Lines 185 and 187: three or two are mentioned, correct wording.

Response: Esteemed reviewer, we have amended the corresponding position of this revised manuscript to "Three physical property indicators, apparent density, water absorption, and the water content of the RCAs, were tested for this study. Moreover, mass variation and carbonation rate, which are two important indicators that reflect the carbonation rate and degree relating to RCAs, were also measured." (Please see first paragraph of section 2.4, highlighted in green). Thank you very much for your constructive and insightful suggestions.

Results and analysis

Redundancy and repetition error is repeated in the text: (when using the concepts increasing, toilet absorption, carbonation ratio) improve wording.

Response: Esteemed reviewer, we feel really sorry for these English language questions, and we have already submitted this revised manuscript to a professional language editing service company (Multidisciplinary Digital Publishing Institute) to help us improve the language of the manuscript. An “Editorial Certificate” was received from “Multidisciplinary Digital Publishing Institute”. This “Editorial Certificate” may certify that this revised manuscript was edited for proper English language, grammar, punctuation, spelling, and overall style by one or more of the highly qualified native English speaking editors at “Multidisciplinary Digital Publishing Institute”. We believe that the language of the edited manuscript should be English-ready for publication.

All the authors of this paper sincerely entreat that you can accept our consideration for this question, thank you very much.

Line 249: pore densification is mentioned, by definition these are the presence of voids in the matrix, so we would speak of pore filling, mass increase, pore dimension reduction?, it is suggested to use a more appropriate technical term that describes the observed phenomenon.

Response: Esteemed reviewer, we feel really sorry about these mistakes and we have already revised the corresponding contents as the “pore filling, mass increase, pore dimension reduction, etc.” in our revised manuscript. Thank you very much for your constructive and insightful suggestions.

The figures must show the error or standard deviation, a statistical analysis of the results is suggested.

Response: Esteemed reviewer, we have added the error bars to the revised figures of our revised manuscript to show the error or standard deviation of the parallel sample statistical results. Thank you very much for your constructive and insightful suggestions.

Lines 325, 326, 333, 334: improve wording.

Response: Esteemed reviewer, we feel really sorry for these English language questions, and we have already submitted this revised manuscript to a professional language editing service company (Multidisciplinary Digital Publishing Institute) to help us improve the language of the manuscript. This may certify that this revised manuscript was edited for proper English language, grammar, punctuation, spelling, and overall style by one or more of the highly qualified native English speaking editors at “Multidisciplinary Digital Publishing Institute”. We believe that the language of the edited manuscript should be English-ready for publication. All the authors of this paper sincerely entreat that you can accept our consideration for this question, thank you very much.

The SEM methodology or the parameters used to make the observations shown are not mentioned, it is necessary to describe it in detail to validate and be able to replicate these results, in the same way the use of an image analysis program that corroborates the results shown by this method is requested.

Response:Esteemed reviewer, “For this paper’s SEM analysis, the RCA samples require to be dried in a vacuum environment at least 3 days. Subsequently, the samples should be coated with gold [1] before SEM test. In this paper, the SEM for RCA samples was performed with a ZEISS Sigma 300, and its work voltage was 15 kV.” The aforementioned contents we have already supplemented in Section 3.7 of our revised manuscript (Please see first paragraph of section 2.2, highlighted in green). All the authors of this paper sincerely entreat that you can accept our consideration for this question, thank you very much.

[1] Y. Song, M. Dong, Z. Wang, X. Qian, D. Yan, S. Shen, L. Zhang, G. Sun, J. Lai, S. Ruan, Effects of red mud on workability and mechanical properties of autoclaved aerated concrete (AAC), J. Build. Eng. 61 (2022) 105238.

Conclusions: ok

Bibliography: ok

See previous comments.

What is said above are the entire responses to all your questions, and we greatly appreciate your thorough review and hope that this revised new manuscript in its present form will be acceptable for publication at Journal of Marine Science and Engineering, thank you and best wishes.

Sincerely yours,

Wei Qin and Xinhui Fan on behalf of all the authors

Round 2

Reviewer 2 Report

Comments and Suggestions for Authors

Thank you for addressing all my previous comments and suggestions. The quality of the manuscript has improved and I recommend it to be accepted for publishing. Congratulations to an interesting study and a good paper!

Reviewer 3 Report

Comments and Suggestions for Authors

The authors have made the minimum modifications necessary to accept this work. It can be published.